# TF-YOLO: An Improved Incremental Network for Real-Time Object Detection

**Wangpeng He \*** , **Zhe Huang, Zhifei Wei, Cheng Li and Baolong Guo**

School of Aerospace Science and Technology, Xidian University, Xi'an 710071, China
\* Correspondence: hewp@xidian.edu.cn; Tel.: +86-158-2930-1428

**Abstract:** In recent years, significant advances have been gained in visual detection, and an abundance of outstanding models have been proposed. However, state-of-the-art object detection networks have some inefficiencies in detecting small targets. They commonly fail to run on portable devices or embedded systems due to their high complexity. In this workpaper, a real-time object detection model, termed as Tiny Fast You Only Look Once (TF-YOLO), is developed to implement in an embedded system. Firstly, the k-means++ algorithm is applied to cluster the dataset, which contributes to more excellent priori boxes of the targets. Secondly, inspired by the multi-scale prediction idea in the Feature Pyramid Networks (FPN) algorithm, the framework in YOLOv3 is effectively improved and optimized, by three scales to detect the earlier extracted features. In this way, the modified network is sensitive for small targets. Experimental results demonstrate that the proposed TF-YOLO method is a smaller, faster and more efficient network model increasing the performance of end-to-end training and real-time object detection for a variety of devices.

**Keywords:** real-time detection; deep learning; small objects; embedded system

---

## 1. Introduction

When people glance at an image, they can immediately know what the objects are, which type of the targets are in the image, and where they are [1]. In order to match the excellent human visual system, fast and accurate object detection represents a significant focus of research in the field of computer vision. Their achievements in this area are used in various applications, such as video surveillance, face recognition, human–computer interaction, and self-driving technology, to name just a few [2]. Robust object detection in a simple environment is relatively easy to achieve, but it is hard to guarantee both speed and accuracy of recognition in practice since the real-world environments may look more complex [3].

Deep learning techniques have been widely employed in the field of object detection during the past decade and have become efficient approaches of extracting features from images. Generic object detection based on deep learning is characterized by two factors: plentiful features and robust feature representation capabilities. They are also combined with traditional hand-crafted features [4]. Existing object detection methods based on deep learning can be generally grouped into two categories, the models based on region proposals and the models based on regression [5].

Currently, classical object detection methods based on region proposals include Region-based Convolutional Neural Networks (R-CNNs) [6], Spatial Pyramid Pooling Networks (SPP-net) [7], Fast R-CNNs [8], Faster R-CNNs [9], and Region-based Fully Convolutional Networks (R-FPN) [10]. However, these approaches fail to achieve real-time speed due to the expensive running process and inefficiency of region propositions. The R-CNNs hypothesizes object locations which depends on region proposal algorithms. Features in the R-CNN are, first, extracted from each candidate region, and then fed into convolutional neural networks. Finally, they are evaluated by Support Vector Machines



(SVM) [11]. The computational cost of R-CNNs is dramatically reduced by sharing convolutions across proposal [12]. Recent advances, for instance, SPP-net and Fast R-CNN tend to reduce the training time. However, time-consuming proposal computation is a bottleneck [13]. To this end, a Region Proposal Network (RPN) is proposed, which shares full-image convolutional features with object detection networks [14], as a kind of fully convolutional network (FCN), which can efficiently predict regions with various scales and aspect ratios. The Faster R-CNN is developed by emerging the RPN and the Fast R-CNN into a single network, in which the convolutional features are shared with the down-stream detection network. It achieves nearly real-time rates by using very deep networks and sharing convolutions at test-time, and it can be trained end-to-end for generating detection proposal [15].

Typical object detection models based on regression are You Only Look Once (YOLO) [1] and Single Shot Multi-box Detector (SSD) [16]. As a single neural network, YOLO is extremely simple, which can concurrently predict bounding boxes coordinates and associated class probabilities. Besides, YOLO frame detection is regarded as a regression problem, and it achieves end-to-end target detection without complex pipeline [1], which results in high efficiency. Furthermore, YOLO achieves a higher mean average precision (mAP) than other real-time systems [17]. The SSD detects objects using a single deep neural network, which only needs input images and ground truth boxes for each object in the training process. Based on a feed-forward convolutional network, SSD generates a fixed-size collection of bounding boxes as well as scores of class probabilities. After a non-maximum suppression step, SSD produces the final detections [18]. In general, the SSD has relatively better accuracy than YOLO. Nevertheless, both YOLO and SSD fail to perform well on small target detection, partly because the target may not have enough information to learn from the structures at the very deep convolutional layers [19].

This paper proposes a multi-scale object detector based on deep convolutional networks, with the aiming at designing an efficient and accurate model to detect objects, which emphasizes on small targets. This smaller, faster and more efficient detector is termed a Tiny Fast YOLO (TF-YOLO). Inspired from the YOLOv3-tiny network, the k-means++ algorithm is an efficient approach of data pre-processing. Taking accuracy and real-time performance into consideration, TF-YOLO uses three scales to detect the previously extracted feature. Experimental results demonstrate that TF-YOLO is not only a cost-efficient solution for practical applications, but also an effective way of improving accuracy of object detection, with a high mean average precision.

The remaining sections of this paper are organized as follows: Section 2 describes the basic object detection methods including a brief introduction about the YOLOv3 algorithm, the Darknet framework, and multi-scale detection. Section 3 introduces the TF-YOLO network which extracts features from connecting multiple layers and adopts a multi-scale prediction framework. Section 4 presents experimental verification to validate the effectiveness of the proposed detection model. Finally, conclusions are summarized in Section 5.

## 2. Related Work

### 2.1. Preliminaries on YOLO

YOLO reframes object detection as a single regression problem, which obtains bounding box coordinates and class probabilities straight from image pixels. As one of other cutting-edge detectors, it has many advantages over the others [20]. Firstly, YOLO is exceedingly fast. On a Titan X GPU, its third version can run at 45 frames per seconds without any batch processing. Secondly, YOLO handles the input as a whole when making decisions [21]. Therefore, the contextual information about classes can be encoded. It is less likely to predict false positives on background. Lastly, YOLO is more applicable to unexpected inputs and new domains, owing to its good generalizable representations.

The separate components of objects detection are unified into a single network in YOLOv3. At the beginning, the input image is divided into an $S \times S$ grid. Then $B$ bounding boxes and confidence

score are defined in each grid cell. Each grid cell predicts *C* conditional probabilities, denoted as $P(Class_i|Object)$. If there is an object in the grid, $P(Object) = 1$. Otherwise, $P(Object) = 0$. The confidence score here refers to the accuracy the box predicts and the probability the objects are contained, which is defined as $P(Object) * IOU_{pred}^{truth}$. Intersection-Over-Union (IOU), here, refers to the intersection area between the predicted bounding box and ground truth box, representing a fraction ranging from 0 to 1. It is noteworthy that conditional class probability ($P(Class_i|Object)$) is quite different from the confidence score ($P(Object) * IOU_{pred}^{truth}$). The former is predicted in each grid, while the other is predicted in each bounding box [1]. Multiply these values in the test process, the class-specific scores in each box is defined in Equation (1). These scores encode the probability of the object appearing in the box, as well as how well the bounding box fits the object.

$$P(Class_i|Object) * P(Object) * IOU_{pred}^{truth} = P(Class_i) * IOU_{pred}^{truth} \tag{1}$$

### 2.2. The Network of Darknet19

YOLOv3 follows the principle of coordinate prediction in YOLOv2. For predicting the categories, multi-label and multi-classification are applied instead of original single-label and multi-classification. Meanwhile, YOLOv3 adopts binary cross entropy loss function instead of multi-class loss function.

On a standard computer with Graphics Processing Unit (GPU), it is easy for YOLOv3 to achieve real-time performance [22]. However, in the miniaturized embedded devices, such as Nvidia SoM, the conventional YOLOv3 algorithm runs slowly. The YOLOv3-tiny network can basically satisfy real-time requirements based on limited hardware resource [23]. Therefore, this paper switches to the YOLOv3-tiny algorithm. The Darknet19 structure of the YOLOv3-tiny network is shown in Table 1, which shows streamlined and enables the YOLOv3-tiny network to achieve the desired effect in miniaturized devices.

**Table 1.** You Only Look One v3-tiny (YOLOv3-tiny) network structure.

| Layer | Type | Filters | Size/Stride | Input | Output |
|:---:|:---:|:---:|:---:|:---:|:---:|
| 0 | Convolutional | 16 | $3 \times 3/1$ | $416 \times 416 \times 3$ | $416 \times 416 \times 16$ |
| 1 | Maxpool | | $2 \times 2/2$ | $416 \times 416 \times 16$ | $208 \times 208 \times 16$ |
| 2 | Convolutional | 32 | $3 \times 3/1$ | $208 \times 208 \times 16$ | $208 \times 208 \times 32$ |
| 3 | Maxpool | | $2 \times 2/2$ | $208 \times 208 \times 32$ | $104 \times 104 \times 32$ |
| 4 | Convolutional | 64 | $3 \times 3/1$ | $104 \times 104 \times 32$ | $104 \times 104 \times 64$ |
| 5 | Maxpool | | $2 \times 2/2$ | $104 \times 104 \times 64$ | $52 \times 52 \times 64$ |
| 6 | Convolutional | 128 | $3 \times 3/1$ | $52 \times 52 \times 64$ | $52 \times 52 \times 128$ |
| 7 | Maxpool | | $2 \times 2/2$ | $52 \times 52 \times 128$ | $26 \times 26 \times 128$ |
| 8 | Convolutional | 256 | $3 \times 3/1$ | $26 \times 26 \times 128$ | $26 \times 26 \times 256$ |
| 9 | Maxpool | | $2 \times 2/2$ | $26 \times 26 \times 256$ | $13 \times 13 \times 256$ |
| 10 | Convolutional | 512 | $3 \times 3/1$ | $13 \times 13 \times 256$ | $13 \times 13 \times 512$ |
| 11 | Maxpool | | $2 \times 2/1$ | $13 \times 13 \times 512$ | $13 \times 13 \times 512$ |
| 12 | Convolutional | 1024 | $3 \times 3/1$ | $13 \times 13 \times 512$ | $13 \times 13 \times 1024$ |
| 13 | Convolutional | 256 | $1 \times 1/1$ | $13 \times 13 \times 1024$ | $13 \times 13 \times 256$ |
| 14 | Convolutional | 512 | $3 \times 3/1$ | $13 \times 13 \times 256$ | $13 \times 13 \times 512$ |
| 15 | Convolutional | 255 | $1 \times 1/1$ | $13 \times 13 \times 512$ | $13 \times 13 \times 255$ |
| 16 | YOLO | | | | |
| 17 | **Route 13** | | | | |
| 18 | Convolutional | 128 | $1 \times 1/1$ | $13 \times 13 \times 256$ | $13 \times 13 \times 128$ |
| 19 | Up-sampling | | $2 \times 2/1$ | $13 \times 13 \times 128$ | $26 \times 26 \times 128$ |
| 20 | **Route 19 8** | | | | |
| 21 | Convolutional | 256 | $3 \times 3/1$ | $13 \times 13 \times 384$ | $13 \times 13 \times 256$ |
| 22 | Convolutional | 255 | $1 \times 1/1$ | $13 \times 13 \times 256$ | $13 \times 13 \times 256$ |
| 23 | YOLO | | | | |

*2.3. Multi-Scale Prediction in Detecting Objects*

Detecting objects at different scales used to be a challenging research topic in the field of computer vision [24]. Feature pyramids built on image pyramids form the fundamentals of a standard solution, partly because of their intensity in computing and their memory. Recently proposed target detectors based on deep learning have avoided pyramid representations [25]. Nevertheless, image pyramids are not the best approach to compute a multi-scale feature representation. In order to naturally leverage the inherent multi-scale and pyramidal shape in the feature hierarchy, the in-network feature pyramids can replace the image pyramids [26] without sacrificing speed and memory.

Relative top-down architectures with skip connections are popular in state-of-the-art object detection research. The YOLOv3-tiny creates a feature pyramid with strong semantics at two scales by adopting subsampling layers and a fusion approach. As shown in Figure 1, the size of the two scales are $13 \times 13$ and $26 \times 26$, which are obtained in the detection of ordinary size target, respectively. Finally, two scales are merged in the end. The architecture is constructed as a feature pyramid, wherein predictions are independently made on each level. The feature pyramid has rich semantics via a top-town pathway and lateral connections [27]. In this way, YOLOv3-tiny has the ability to detect small targets.

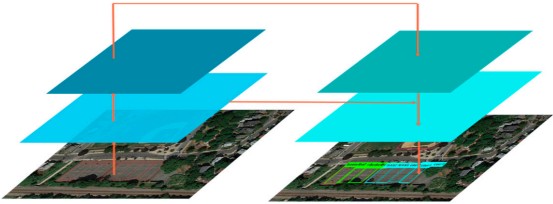

**Figure 1.** Multi-scale prediction in the YOLOv3-tiny network. There are two scales prediction in the YOLOv3-tiny network. The sizes of the tensors are $13 \times 13$ and $26 \times 26$, respectively.

## 3. Proposed TF-YOLO Network

Previous methods, such as SSD and YOLOv3, regard detection as a regression problem, which have successfully achieved efficient and accurate results. Nevertheless, these methods fail to detect objects on an embedded system in real-time. This section introduces the proposed Tiny Fast YOLO (TF-YOLO) network in detail. As shown in Figure 2, the proposed TF-YOLO network is designed based on the YOLOv3-tiny algorithm, and it attempts to process more efficiently on the above devices. Owning to multi-scale fusion, multi-scale detection, and k-means++ clustering, the TF-YOLO network enables end-to-end training and real-time speeds while keeping high average precision. Therefore, TF-YOLO network performs well on detecting multi-scale targets, especially on recognizing smaller targets. The framework flowchart of TF-YOLO network is shown in Figure 3.

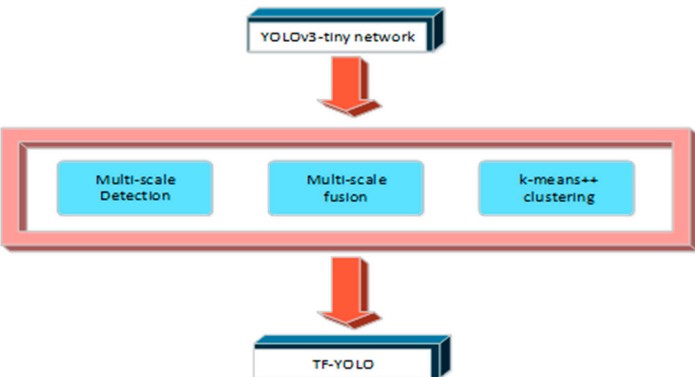

**Figure 2.** The designed concept of the proposed tiny fast You Only Look Once (TF-YOLO) network. Multi-scale detection is more sensitive to small targets. Multi-scale fusion makes full use of features of the input image. k-means++ is applied to find more excellent priori boxes of the targets.

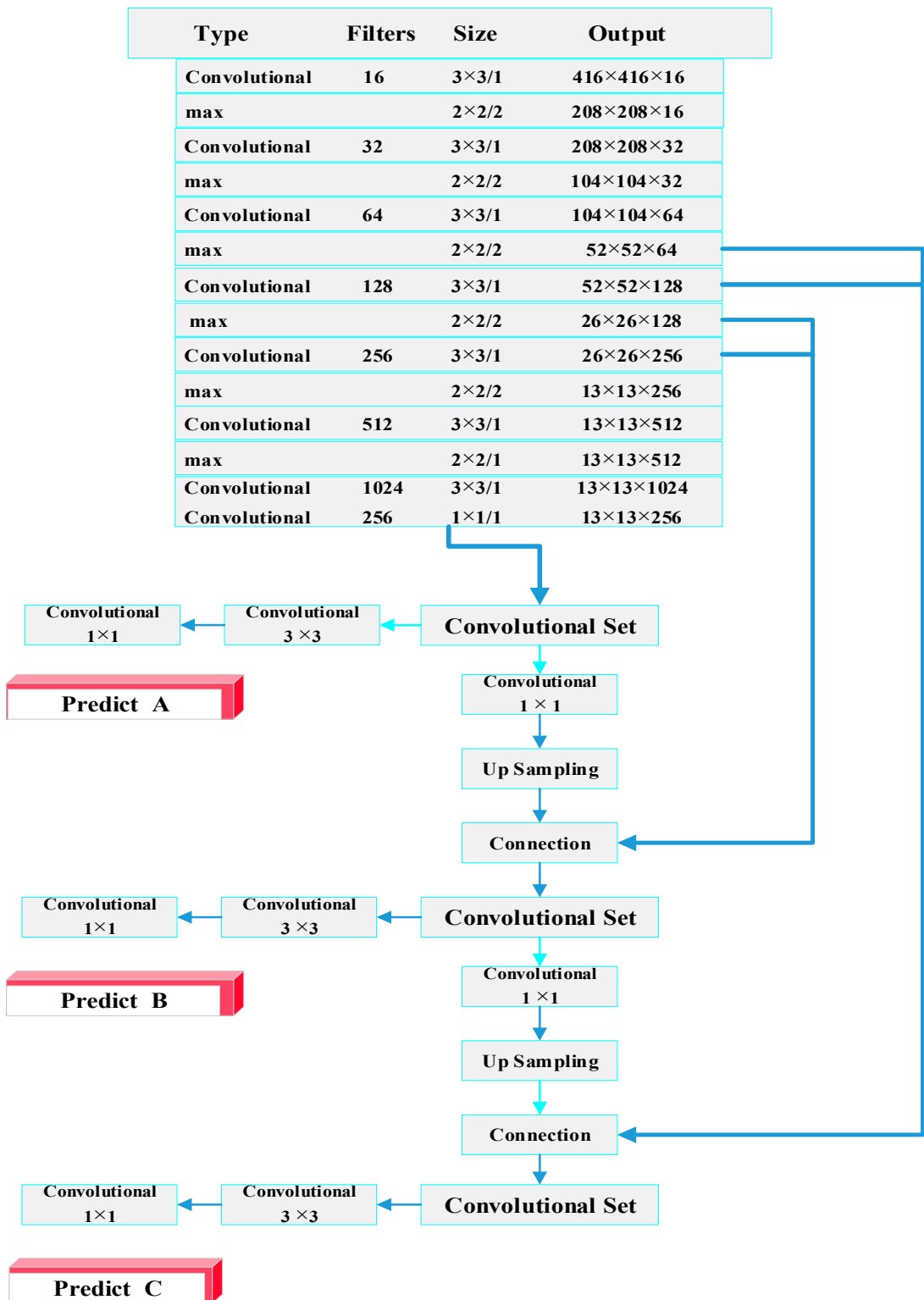

| Type | Filters | Size | Output |
|------|---------|------|--------|
| Convolutional | 16 | 3×3/1 | 416×416×16 |
| max | | 2×2/2 | 208×208×16 |
| Convolutional | 32 | 3×3/1 | 208×208×32 |
| max | | 2×2/2 | 104×104×32 |
| Convolutional | 64 | 3×3/1 | 104×104×64 |
| max | | 2×2/2 | 52×52×64 |
| Convolutional | 128 | 3×3/1 | 52×52×128 |
| max | | 2×2/2 | 26×26×128 |
| Convolutional | 256 | 3×3/1 | 26×26×256 |
| max | | 2×2/2 | 13×13×256 |
| Convolutional | 512 | 3×3/1 | 13×13×512 |
| max | | 2×2/1 | 13×13×512 |
| Convolutional | 1024 | 3×3/1 | 13×13×1024 |
| Convolutional | 256 | 1×1/1 | 13×13×256 |

**Figure 3.** Workflow of the proposed TF-YOLO network.

### 3.1. The Features of Multiple Layers Concatenation

When deep neural networks start converging, the degradation problem will be exposed, and the accuracy will deteriorate rapidly in the end. Aiming to address that problem, this paper follows DenseNet proposed by Huang et al. [2]. Short connections in DenseNet enable the training process to be easier and more accurate in CNN, which is of great importance in image classification. In the

Dense block, previous convolutional layer output is transferred to the subsequent one. Hence, more complicated features are extracted by the filters of the convolutional layers, which are included by multi-layer networks. It can also be understood that all layers are directly connected with each other, thus apparently alleviating gradient disappearance. By incorporating those ideas into the hidden network of DenseNet, the sufficient features of the above network can be extracted. Since the layer is not very deep, the output layer can extract the previous features. Subsequently all of them have been connected together. This paper adopts the principle of multiple layers concatenation and eventually achieves satisfactory performance, which will be thoroughly presented in Section 4.

The detailed all-layer connection method utilized in the proposed TF-YOLO detection approach is display in Figure 4. Specifically, the tenth, eleventh and thirteenth layers of the designed network are connected, and then these layers feed into the convolutional layer followed by the up-sampling layer. Similarly, the tensors in eighth and ninth layers are processed together, forming an innovative input and entering into the next layer. Finally, the corresponding in sixth and seventh layers and the tensor are connected to the next hidden layer.

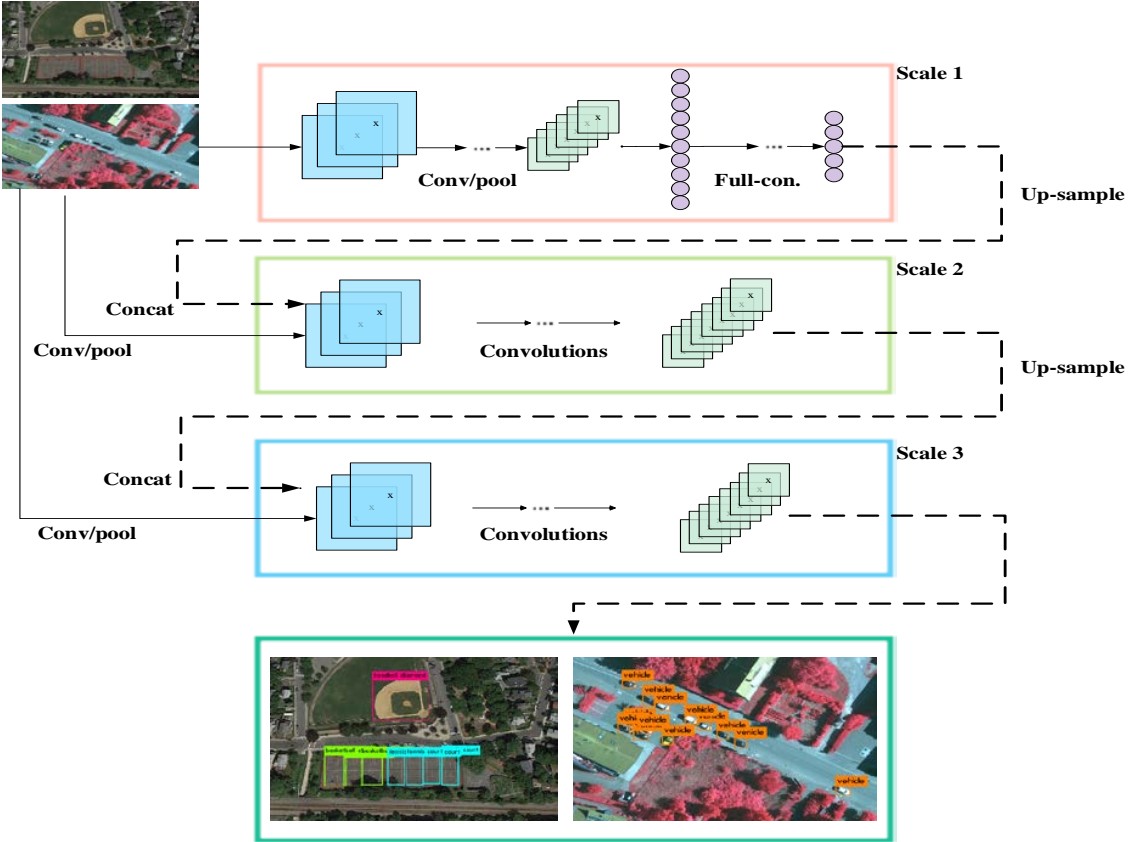

**Figure 4.** The workflow of connecting multiple layers.

### 3.2. Multi-Scale Prediction Framework

Small target detection is a significant challenge that commonly causes traditional detectors to fail. Multi-scale prediction is an important step towards understanding and inferring different objects, especially small targets, and their arrangements observed in a scene. This section presents an improved FPN-based multi-scale prediction framework and integrates it to a particular filter detector to address that problem. The major advantage of FPNs is that they produce a multi-scale feature representation in which all levels are semantically strong. As a result, predictions can be made on the finest level. In addition, it can be trained end-to-end with three scales and be used consistently during training and testing process. Therefore, FPNs are able to achieve higher accuracy without increasing testing time over the single-scale baseline.

In the proposed work, several convolutional layers are added from the feature extractor. The last three predict a three-dimensional tensor encoding bounding box coordinates, object prediction, and class predictions [1]. Assuming that there are 10 classes in the experiment, and three boxes at each scale are predicted. Thus, the tensor is $M \times M \times [3 \times (4 + 1 + 10)]$, for four bounding box offsets, one object prediction, and ten class predictions.

The proposed TF-YOLO network adopts the advances of the Darknet structure. The neural network is not deep, whereas, the features in different scales are merged in the proposed TF-YOLO network. It connects the feature maps with the same feature scale in the above mentioned Darknet structure. Meanwhile, the network extracts the feature map from two former convolution layers, followed by the up-sampling layer. Then the tensors above are connected together. In this way, the characteristics of the hidden layer, as well as the deep features can be extracted by the full-connection layer. Section 3.1 explains how to make hierarchical connections in detail. In the first layer, the size of the tensor is $13 \times 13 \times 18$. Via two convolutional layers and an up-sampling layer, the tensor becomes $26 \times 26 \times 18$, which predicts the second scale. This procedure repeats one more time, and the tensors becomes $52 \times 52 \times 18$.

In the proposed TF-YOLO network, these three scales are used to detect targets in various sizes. The network performs large-scale detection in $13 \times 13$ size map and detects the moderate-scale target in $26 \times 26$. The small target is detected in $52 \times 52$ size map. By connecting the multiple features of the same scale, the TF-YOLO network prominently promotes the ability to detect objects. The feature extraction workflow of the TF-YOLO network is shown in Figure 5.

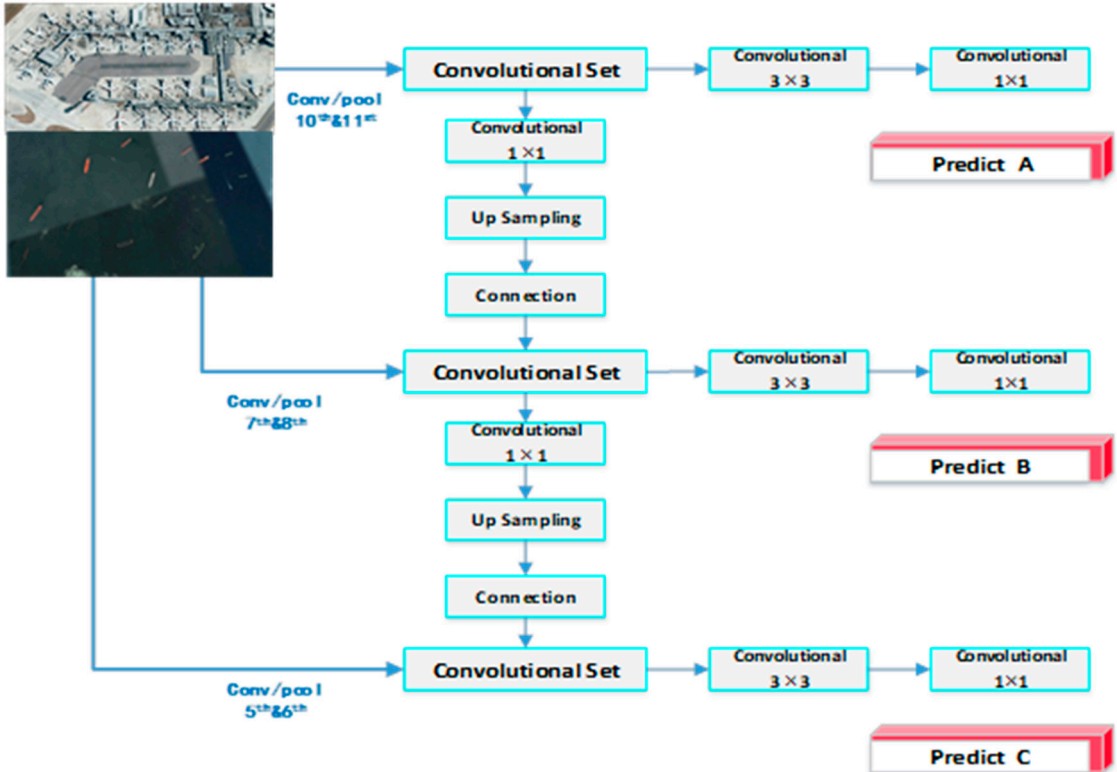

**Figure 5.** Feature extraction diagram of the proposed TF-YOLO network.

### 3.3. K-means++ Clustering in Pre-Processing

Clustering algorithms is generally viewed as an unsupervised method for data analysis [28]. K-means++ clustering is an approach commonly used to adaptively partition a dataset into groups. It is necessary to specify the number of cluster centers in advance, since $k$ clustering centers are simultaneously initialized. The way to choose the cluster center is to select the maximum or minimum

value randomly in the dataset as the initial cluster center. Therefore, the better the cluster centers are selected, the more effective the algorithm would be obtained.

As mentioned above, K-means++ algorithm is used to conduct latitude clustering at first. It randomly selects a sample point in the dataset as the first initialized cluster center, and it calculates the rest for each cluster center [29]. After the sample points are initialized, the distances between the cluster centers are calculated, then the shortest distance from its own cluster center is chosen. On the other hand, the sample with the greatest distance is selected as the new cluster center. The above process is repeated until there is no further change in the assignment of distances to clusters. Thus, the algorithm converges. Eventually, the final cluster center is calculated using the k-means++ algorithm to determine the specific parameters of the anchor.

The k-means++ algorithm generally uses the Euclidean distance to measure the distance between two points. Nevertheless, there are the following three scale targets in the dataset: large-scale targets, moderate-scale targets, and small-scale targets [30]. The main steps of the K-means++ method are as follows:

- Step 1: Choose an initial center $t_1$ uniformly at random from the dataset $X$.
- Step 2: Choose the next center $t_i$, selecting $t_i = x' \in X$ with probability $\frac{D(x')^2}{\sum_{x \in X} D(x)^2}$, where D(x) is the distance from a data point $x$ to the closest center.
- Step 3: Repeat Step 2, until $D(x)$ is the shortest distance. After which, $k$ center is chosen.
- Step 4: Define center $T = \{t_1, t_2, \cdots, t_k\}$.
- Step 5: For $i \in \{1, 2, \cdots, k\}$, set the cluster $T_i$ to be the set of points in $X$ that are closer to $t_i$ than they are to $t_j$ for all $j \neq i$.
- Step 6: For $i \in \{1, 2, \cdots, k\}$, set $t_i$ to be the center of mass of all points in $T_i$: $t_i = \frac{1}{|T_i|} \sum_{x \in T_i} x$.
- Step 7: Repeat Step 5 and Step 6 until $T$ converges.

The definition of the three sizes of targets here refers to the proportion that they are in the entire image. Furthermore, in terms of the Euclidean distance, more errors would occur in the larger bounding boxes rather than in smaller bounding boxes. Since the goal is to get better IOU through anchor boxes, Jake's distance is a better choice. Jake's distance is adapted to the variable box size, which is a good solution to resolve the error caused by Euclidean distance. The distance formula is defined as:

$$d(box, centroid) = 1 - \text{IOU}(box, centroid) \tag{2}$$

where *box* represents the sample, and *centroid* represents the center of the cluster, and $\text{IOU}(box, centroid)$ represents the intersection of the cluster's center box and the cluster box [31]. The intersection ratio IOU can indicate the accuracy of the prediction box by Equation (3).

$$\text{IOU}(bb_{gt}, bb_{dt}) = \frac{bb_{gt} \cap bb_{dt}}{bb_{gt} \cup bb_{dt}} \tag{3}$$

where $bb_{gt}$ represents the real box, and $bb_{dt}$ represents the prediction box. Combining the above two equations, the final distance can be calculated as

$$d(box, centroid) = \frac{bb_{gt} \cup bb_{dt} - bb_{gt} \cap bb_{dt}}{bb_{gt} \cup bb_{dt}} \tag{4}$$

As mentioned above, YOLOv3 algorithm has achieved end-to-end training and high-speed target detection. However, some problems still exist. Conventional YOLO divides each image into a fixed grid, which results in the number of detected objects will be limited. The fixed parameters provided by anchor are suitable for the targets in the VOC datasets, while they are not adapted to the targets in specific scenes. Common targets, such as vehicles, tanks, and airplanes, have a large aspect ratio. Therefore, this section takes advantage of the ideas in Fast R-CNN and SSD to re-cluster according to a

spacific scenario. In the beginning, the network manually sets a priori box. To guarantees the selection of network is more subjective, and it would make the deep network easier to learn. Furthermore, its predictions perform better than the original method. In order to optimize the adapted parameters and select appropriate anchor box size, the designed network needs to re-cluster according to real application domains. In this way, the TF-YOLO network performs well on multi-scale prediction, meanwhile, it is insensitive to small objects particularly.

## 4. Experimental Verification and Result Analysis

In this section, the performance of the proposed TF-YOLO network is evaluated. Specifically, aerial remote sensing images from NWPU VHR-10 [32–34] are used for training and testing in the experiment. The NWPU VHR-10 dataset contains a total of 800 very-high-resolution (VHR) remote sensing images cropped from Google Earth and the Vaihingen dataset, and manually annotated by experts. The differences between remote sensing and conventional natural images can be briefly described as follows. First of all, there are numerous small targets with a little visual information in the remote sensing images. Assuming that the CNN's pooling layer further reduces the amount of information to the small targets. After four pooling layers, a 24 × 24 target has only approximately 1 pixel, leading to the dimension too diminutive to distinguish. Secondly, there are various scale and perspective of remote sensing images. The perspectives of aerial remote sensing images are basically high-altitude, and the targets on the ground may have different size and mode. Some detectors well trained on conventional datasets may fail to perform well in remotely sensed images. Thirdly, the background of remote sensing images is complex for the large field of view. What's more, a variety of backgrounds will extend a certain amount of interference on testing targets.

### 4.1. Comparison of Speed and Precision

For this test, a total of 500 pictures containing 10 types of objects were selected in NWPU VHR-10 dataset. The detailed 10 categories are airplane, ship, storage tank, baseball diamond, tennis court, basketball court, ground track field, harbor, bridge, and vehicle. Cross validation is used to evaluate the precision of TF-YOLO. During the process of data pre-processing, nine cluster centers are selected, and the targets are distributed into three scales by K-means++. The size of these nine scales are arranged from small to large as follows: (22,19), (46,29), (39,54), (86,52), (71,108), (139,86), (106,161), (231,130), and (289,188). Meanwhile, a total of 300 images containing small and dense objects were selected in VOC 2007 dataset.

The samples, based on their accurate category and prediction class, can be divided into the following four categories [35]: TP (true positive), FP (fault positive), TN (true negative), and FN (fault negative). Precision refers to the proportion of TP in the predicted positive example, and recall refers to the proportion of TP in the truly positive example. They can indicate the number of their corresponding samples. The accuracy and recall rate are defined as follows:

$$\text{precision} = \frac{TP}{TP + FP} \tag{5}$$

$$\text{recall} = \frac{TP}{TP + FN} \tag{6}$$

Generally, the trade-off between accuracy and recall is a tricky problem. In order to evaluate the precision among different types of targets, mAP is introduced, which is one of significant measure metrics to evaluate the test results [36].

Meanwhile, in order to maintain the consistency of the data distribution in each subset, the feature can be extracted through hierarchical sampling layers. AP and mAP of the three sets of comparative experiments are illustrated in Table 2. The first set of comparison experiments is the YOLOv3-tiny network. Inspired by YOLOv3-tiny, the second set of comparison experiments is defined as the YOLO_k network. Without data pre-processing, its structure is the same as TF-YOLO. The third group

of comparative experiments is the TF-YOLO network, which improves the size of anchors through K-means clustering.

**Table 2.** Comparison of the accuracy of the three networks.

| Precision | Classes | YOLOv3-tiny | YOLO_k | TF-YOLO |
|---|---|---|---|---|
| | airplane | 0.81806 | 0.81926 | **0.85771** |
| | ship | 0.89595 | 0.92718 | **0.99084** |
| | storage tank | 0.74237 | 0.76223 | **0.77361** |
| | baseball diamond | 0.73624 | 0.82734 | **0.84513** |
| **AP(%)** | tennis court | 0.88762 | 0.89091 | **0.89542** |
| | basketball court | 0.91849 | 0.93965 | **0.95219** |
| | ground track field | 1.0 | 1.0 | **1.0** |
| | harbor | 0.92718 | 0.94183 | **1.0** |
| | bridge | 0.75806 | 0.79562 | **0.81266** |
| | vehicle | 0.68493 | 0.73731 | **0.81731** |
| **mAP(%)** | | 0.83689 | 0.86413 | **0.89449** |

One significant advantage of the proposed TF-YOLO network is real-time working on portable devices. For instance, the TF-YOLO network can be applied to the embedded system on the Nvidia Jetson TX2. After training models in NWPU VHR-10, the TF-YOLO network runs at about 24.3 FPS. However, the YOLOv3-tiny network runs at 24.6 FPS. In embedded systems, the YOLOv3-tiny network is slightly faster than the TF-YOLO network. Partly because the network in TF-YOLO is deeper than YOLOv3-tiny, and more parameters in TF-YOLO are learnt during the training period. However, the accurate of the TF-YOLO network is significantly improved than the YOLOv3-tiny network, which results in the TF-YOLO network performing is well on detecting multi-scale targets with real-time speed.

As obviously revealed in Table 2, the mAPs of TF-YOLO network are prominently higher than those of the YOLOv3-tiny and YOLO_k networks, regardless of whether complex objects are included. The YOLOv3-tiny network does not respond satisfactory to the targets in testing images. Besides, by improving the network structure in the YOLO_k network, mAP significantly improves to 0.86413, but it is still unsatisfactory. On the other hand, the proposed TF-YOLO network, within K-means++ clustering to change anchors, demonstrates the highest AP of the single-class target and achieves 0.89449 in mAP.

In VOC 2007 dataset, a total of 200 images containing small targets were selected. In consideration of the detection results, the TF-YOLO and YOLOv3-tiny networks were chosen for comparison with state-of-art methods based on region proposals, including SPP-net, RCNN, Faster RCNN, and YOLOv3. The APs and mAPs of the above methods are displayed in Table 3.

**Table 3.** Small objects detection results for region-based proposal methods on VOC 2007 dataset.

| Method | mAP (%) | AP (%) | | | | | | | | | |
|---|---|---|---|---|---|---|---|---|---|---|---|
| | | Aero | Bird | Boat | Car | Chair | Dog | Person | Plant | Sheep | Cow |
| SPP-net | 30.3 | 42.7 | 33.9 | 27.5 | 24.8 | 25.3 | 11.2 | 34.2 | 15.1 | 41.6 | 43.7 |
| RCNN | 36.2 | 44.9. | 38.2 | 23.4 | 38.6 | 29.3 | 15.2 | 37.6 | 19.7 | 46.5 | 68.6 |
| Faster RCNN | 67.9 | 74.0 | 58.7 | 66.3 | 72.5 | 45.7 | 69.5 | 73.6 | 56.7 | 86.4 | 75.7 |
| YOLOv3 | 55.9 | 68.5 | 41.2 | 50.4 | 80.3 | 57.9 | 68.5 | 36.7 | 32.6 | 51.6 | 71.4 |
| YOLOv3-tiny | 27.2 | 39.9 | 20.5 | 12.9 | 33.6 | 18.7 | 11.4 | 23.4 | 15.3 | 41.7 | 54.7 |
| **TF-YOLO** | **31.5** | **42.6** | **35.4** | **19.7** | **35.4** | **22.1** | **12.7** | **29.7** | **15.7** | **42.1** | **59.2** |

Table 3 shows that the AP scores of the TF-YOLO method are higher than those of the classical methods in every object class. When small objects are included in complicated background, the mAP of the TF-YOLO method is 31.5%, which is higher than that of the YOLOv3-tiny method by 27.2%, and SPP-net by 30.3%. When compared with YOLOv3, RCNN, and Faster RCNN, the precision of the

TF-YOLO method is lower than these methods. Nevertheless, the detection speed of the TF-YOLO method is much faster than these classical methods, which is shown in Table 4. Taking both accuracy and detection speed into consideration, TF-YOLO method exhibits the best performance in small object detection among the above methods in an embedded system.

**Table 4.** Comparison of precision and speed for region-based proposal methods on VOC 2007 dataset.

| Method | SPP-net | RCNN | Faster RCNN | YOLOv3 | YOLOv3-tiny | TF-YOLO |
|---|---|---|---|---|---|---|
| mAP (%) | 30.3 | 36.2 | 67.9 | 55.9 | 27.2 | **31.5** |
| Run time (sec/img) | 0.38 | 0.82 | 0.26 | 0.13 | 0.10 | **0.09** |

*4.2. Comparison of Loss Curves*

As revealed in Figure 6, the loss curve of the TF-YOLO network converges faster than the YOLOv3-tiny network. Specifically, the loss curve of the YOLOv3-tiny network converges approximately from 0.1. Whereas, the loss curve of TF-YOLO starts to converge approximately from 0.05.

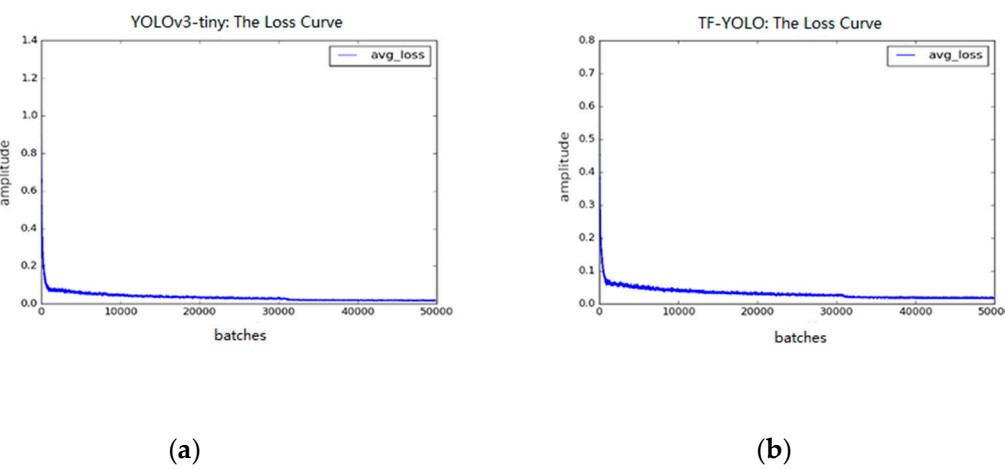

(**a**)                                                      (**b**)

**Figure 6.** The comparison of loss curves for two networks: (**a**) YOLOv3-tiny network; (**b**) TF-YOLO network.

*4.3. Comparison of IOU Curve*

IOU is the intersection over union of predicted bounding box and ground truth. The ideal situation is complete overlap, and IOU should approach 1. In general, if IOU > 0.7, it can be considered as a good result [37]. For the loss function of the training model, the sum-squared error is used to integrate the localization error (bounding boxes coordinate error) and the classification error. Figure 7 shows the IOU curve of the YOLOv3-tiny network and the loss curve of the TF-YOLO network, respectively.

Compared with the IOU curve of the YOLOv3-tiny network, the area under the curve of the TF-YOLO network is larger. Furthermore, the IOU curve of the TF-YOLO network converges faster. The TF-YOLO network achieves a higher overlap between the candidate bound and the ground truth bound, which means the ratio of their intersection to union is greater, which indicates that the predicted bounding box is close to the ground truth. In summary, the performance of the TF-YOLO network has been greatly improved.

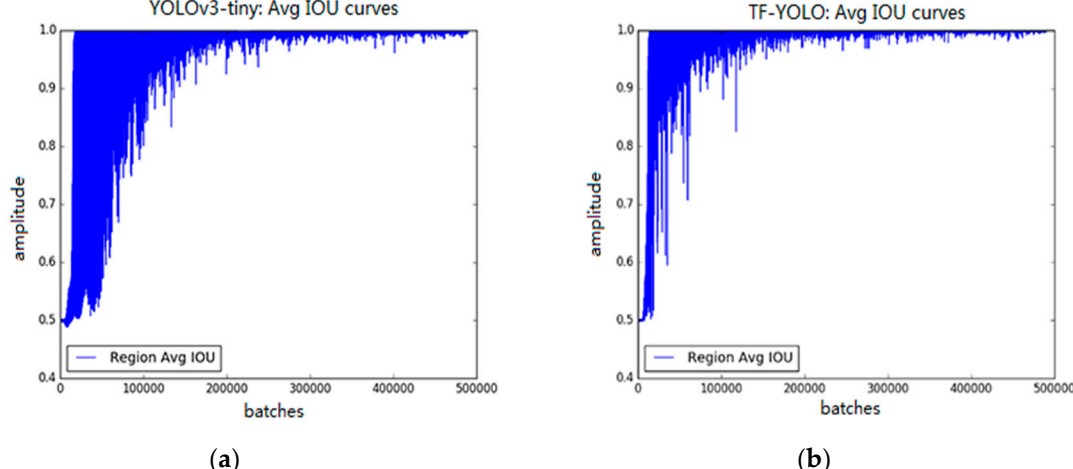

**Figure 7.** The comparison of intersection over union (IOU) curves obtained by two networks: (**a**) YOLOv3-tiny network; (**b**) TF-YOLO network.

*4.4. Qualitative Analysis*

In aerial photography and remote sensing images, the targets are prone to be smaller in size and are under complicated background. Sometimes, the scale and orientation would be various. Therefore, the conventional algorithms generally fail to detect these targets.

In this section, the YOLOv3-tiny network is chosen for comparison with the proposed TF-YOLO network. Figure 8 depicts a total of 16 pictures of 8 scenarios. The pictures in the first and third rows are performed by the YOLOv3-tiny network. For comparison, the pictures in the second and fourth rows are detected by the proposed TF-YOLO network. It is easy to learn that there is certain missed and false detection in the YOLOv3-tiny network. In contrast, almost all visible targets are effectively monitored by the TF-YOLO network. For a clearer representation, the above corresponds to the same picture. It is easy to see that there are certain missed and false detection in the YOLOv3-tiny network. On the other hand, TF-YOLO network has a much better detection effect, and almost all the targets to be inspected are detected.

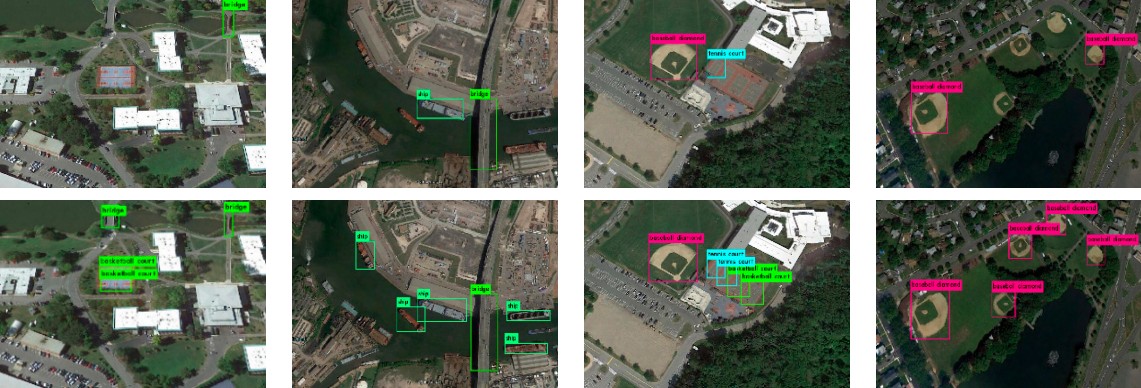

**Figure 8.** *Cont.*

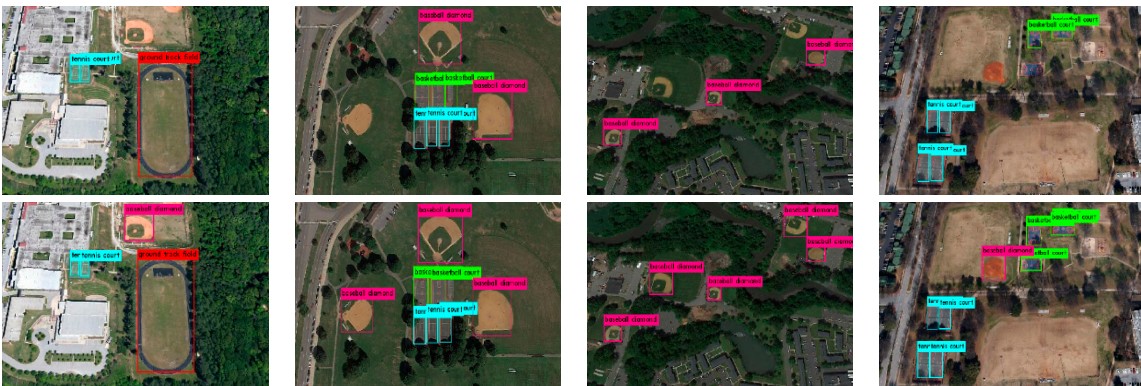

**Figure 8.** Visual comparisons of detection results between the classical YOLOv3-tiny method (the first and third rows) and the TF-YOLO method (the second and fourth rows).

To guarantee the objectiveness of evaluating the performance of the proposed TF-YOLO network, detection results of 20 randomly selected test images from the test-set are also illustrated. The testing results are shown in Figure 9. Experimental results indicate that the proposed TF-YOLO network enables a better retrieval capability and a higher detection accuracy for object detection, and it is sensitive for small targets.

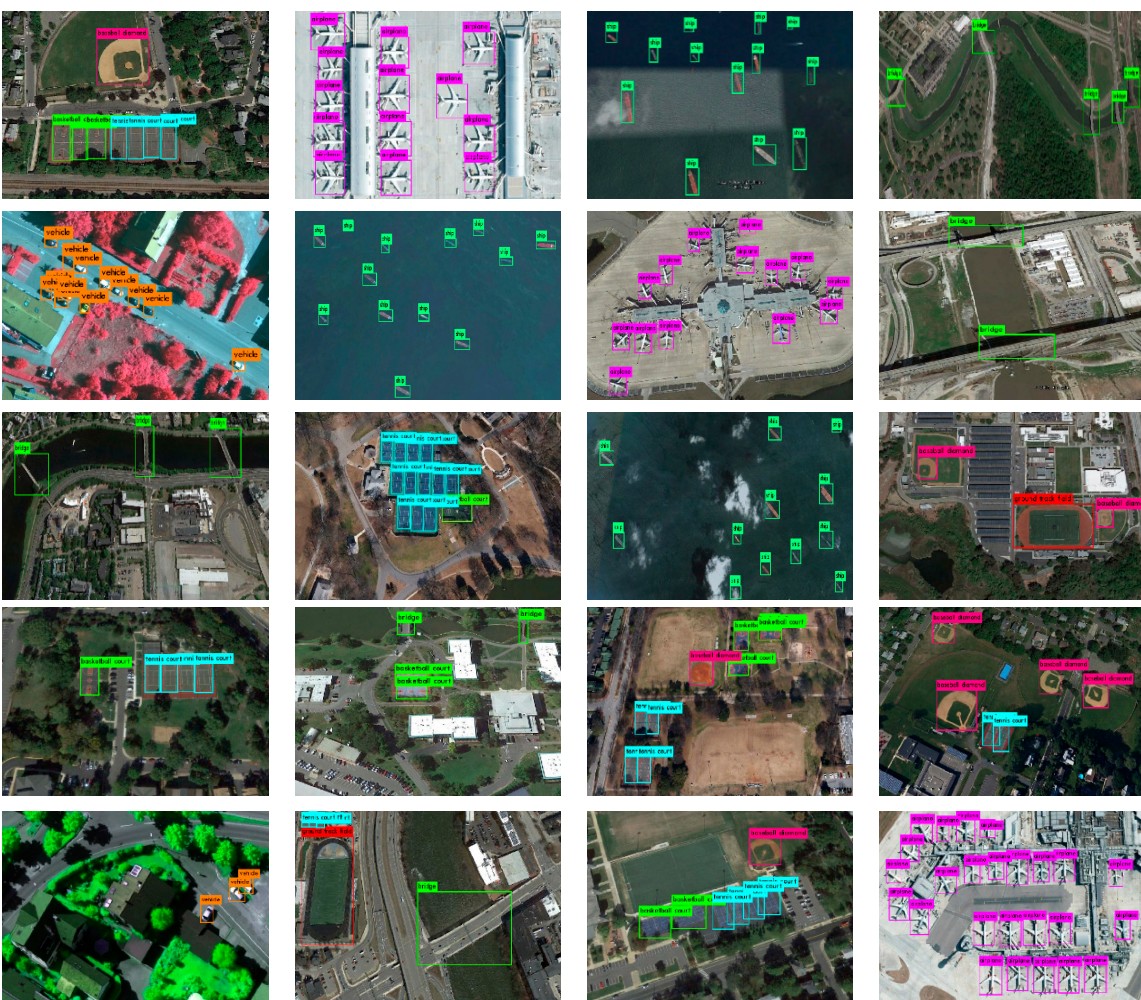

**Figure 9.** The test results in the proposed TF-YOLO network.

## 5. Conclusions

This paper proposes a multi-scale object detection approach for small targets detection. The improved incremental network for real-time object detection is termed as Tiny Fast YOLO (TF-YOLO) network. The major improvement of the TF-YOLO network is owning to its structural optimization of the YOLOv3-tiny network. In addition, introducing the K-means++ algorithm as a starting point, the TF-YOLO network can get a better priori box for each target, with clustering the dataset and selecting the number and specifications of the candidate frames. In this way, the TF-YOLO network can carry out multi-scale prediction, and the accuracy of the detection of small targets has also been significantly improved. Compare to conventional detectors, this paper is a smaller, faster and more efficient detector, increasing the performance of end-to-end training and real-time object detection to a variety of devices, even the embedded systems or portable devices. Experimental results demonstrate that the TF-YOLO network takes full advantage of image features in the framework and improves the performance on small targets with less time consuming. Considering a trade-off between accuracy and speed, the proposed TF-YOLO network exhibits the best performance in small object detection among the state-of-the-art methods. In future work, the lower level features will be richly extracted with the multi-scale paradigm to promote detection performance.

**Author Contributions:** All the authors contributed to this study. W.H. conceptualization, funding acquisition, project administration and editing; Z.H. investigation, writing of the original draft, designing the network and experiments; Z.W. and C.L. analyzing the data and investigation; B.G. supervision.

**Funding:** The authors would like to thank the editor and anonymous reviewers for their valuable comments on this paper. This research is supported financially by the National Natural Science Foundation of China (Grant No. 51805398), the National Natural Science Foundation of Shaanxi Province (Grant No. 2018JQ5106), and the Fundamental Research Funds for the Central Universities (Grant No. JBX171308).

**Conflicts of Interest:** The authors declare no conflict of interest.

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
