# Peer review of "TF-YOLO: An Improved Incremental Network for Real-Time Object Detection"

_applsci, doi:10.3390/app9163225_

Round 1

Reviewer 1 Report

Unfortunately I can't recommend after seeing Table 3 and Table 4 being just 3 times faster and half accurate than RCNN is not enough. If the precision was close to RCNN with this speed it could justify publication.

Author Response

Comments:

Unfortunately, I can't recommend after seeing Table 3 and Table 4 being just 3 times faster and half accurate than RCNN is not enough. If the precision was close to RCNN with this speed it could justify publication.

Authors:

The authors thank the reviewer for his/her valuable comments and suggestions. Those comments are all constructive and very helpful for revising and improving our paper.

In this experiment, we select a total of 200 images containing small targets in VOC 2007 dataset. TF-YOLO and YOLOv3-tiny were chosen for comparison with state-of-art methods based on region proposals, including SPP-net, Faster RCNN and YOLOv3. As shown in Table3 and Table4, the proposed TF-YOLO is 3 times faster and half accurate than Faster RCNN.

In addition, we added the experiments and compared with RCNN method in the same dataset. When compared with YOLOv3, RCNN and Faster RCNN respectively, the precision of TF-YOLO is lower than above methods. Nevertheless, the detection speed of TF-YOLO is much faster than state-of-the-art methods. Specifically, the accuracy of TF-YOLO is close to RCNN while significantly improving time efficiency. When making a trade-off between accuracy and speed, TF-YOLO exhibits the best performance in small object detection among the above methods.

Table 3. Small objects detection results for region-proposed-based methods on VOC 2007 dataset.

Method

mAP

(%)

AP (%)

Aero

Bird

Boat

Car

Chair

Dog

Person

Plant

Sheep

Cow

SPP-net

30.3

42.7

33.9

27.5

24.8

25.3

11.2

34.2

15.1

41.6

43.7

RCNN

36.2

44.9.

38.2

23.4

38.6

29.3

15.2

37.6

19.7

46.5

68.6

Faster RCNN

67.9

74.0

58.7

66.3

72.5

45.7

69.5

73.6

56.7

86.4

75.7

YOLOv3

55.9

68.5

41.2

50.4

80.3

57.9

68.5

36.7

32.6

51.6

71.4

YOLOv3-tiny

27.2

39.9

20.5

12.9

33.6

18.7

11.4

23.4

15.3

41.7

54.7

TF-YOLO

31.5

42.6

35.4

19.7

35.4

22.1

12.7

29.7

15.7

42.1

59.2

                    Table 4. Comparison of precision and speed for region-proposed-based methods on VOC 2007 dataset.

Method

SPP-net

RCNN

Faster RCNN

YOLOv3

YOLOv3-tiny

TF-YOLO

mAP (%)

30.3

36.2

67.9

55.9

27.2

31.5

Run time (sec/img)

0.38

0.82

0.26

0.13

0.10

0.09

Reviewer 2 Report

I believe the authors have made a good job in revisiting the manuscript, which improved considerably. 

Minor issues: 

- pre-procession -> pre-processing

- please, avoid widow titles such as 2.3

Author Response

Comments and Suggestions for Authors:

I believe the authors have made a good job in revisiting the manuscript, which improved considerably.

Minor issues:

- pre-procession -> pre-processing

- please, avoid widow titles such as 2.3

Authors:

The authors thank the reviewer for his/her valuable comments and suggestions. Those comments are all constructive and very helpful for revising and improving our paper. We really appreciate that the reviewer commented on our proposed approach that “authors have a good job in revisiting the manuscripts”. Meanwhile, we have considered the valuable comments raised by the reviewer and made changes accordingly in the revised manuscript. Note that changes are marked with blue in the revised manuscript.

The main modifications can be summarized as follows:

(1) We have rechecked the manuscript to proofread our work. Some grammatical mistakes, typos and inappropriate sentences have been corrected or rewritten in the revised manuscript.

(2) Some inappropriate expressions have been adjusted, which would be more objective and precise, such as “pre-procession”.

(3) We have revised some sub-titles, which enables this paper easier for readers to understand. Accordingly, the contexts related to all figures has also been changed in the manuscript to better present the contribution and main goal of our paper.

Again, we would like to express our sincere thanks to the reviewer for your constructive comments and suggestions.

Round 2

Reviewer 1 Report

Fine by me because it is stressed to run on systems of low computational power.

This manuscript is a resubmission of an earlier submission. The following is a list of the peer review reports and author responses from that submission.

Round 1

Reviewer 1 Report

This paper presents a method to more accurately detect small objects in images.

It is not a surprise that the method would do more up-scaling to achieve better accuracy than the previous literature. 

Also suppose that the specific paper is used from the military and want to see if it will neutralize an enemy battle ship or a fishing boat. So I am deeply interested in the precision and not the mean precision. By taking the mean when you go small you take away the noise while I am really interested about the noise. 

So. Do you have more False Positives than the other methodology? Please provide experimentation on this issue also.

Also please kindly look again Equation (1) It looks fishy. 

Now we come at the language, I am sorry to tell it needs a lot of attention.

All in all major revisions needed.

Reviewer 2 Report

This submission proposes TF-YOLO, a tiny fast YOLO model conceived for embedded system applications. While the idea is interesting and has potential, in my opinion the authors failed to communicate it in a proper manner. The manuscript should have been out proofread by a more fluent English speaker: it contains a lot of mistakes and typos and this does not allow the reader to appreciate the contribution of the paper. In fact, the core of the paper, i.e. the proposed method, is not clearly described, and it is difficult to understand how it works.

There are problems also with the structure; for example, the dataset is described after the experimental results have been provided. Some parts are very didactic and thus unessential, for example the description of the k-means++ algorithm or the formulation of the cross-validation procedure.

Methodologically, the chosen dataset is really small, and this does not allow one to claim the superiority of one model over the other in terms of accuracy of whatever. In addition, the choice of the size of the nine scales appears to be somewhat arbitrary without an underlying rationale.

Finally, the manuscript has not been carefully read before submission, due to the presence of the above-mentioned typos and of the sentence “The higher the relevant document rank, the better the accuracy is”, which refers to a completely different context.